# Myxofibrosarcoma of the Chest Wall Detected on ^99m^Tc-MDP Whole-Body Bone Scan

**DOI:** 10.3390/diagnostics14080857

**Published:** 2024-04-22

**Authors:** Chia-Hsuan Lee, Hueng-Yuan (Daniel) Shen, Yow-Ling (Shirley) Shiue, Hung-Yen Chan, Hung-Pin Chan

**Affiliations:** 1Department of Nuclear Medicine, Kaohsiung Veterans General Hospital, Kaohsiung City 813, Taiwan; brightha7@gmail.com (C.-H.L.); hyshen@vghks.gov.tw (H.-Y.S.); hongyenchan0407@yahoo.com.tw (H.-Y.C.); 2Department of Psychiatry, Kaohsiung Medical University Hospital, Kaohsiung City 807, Taiwan; 3Institute of Biomedical Sciences, College of Medicine, National Sun Yat-sen University, Kaohsiung City 804, Taiwan; shirley@imst.nsysu.edu.tw; 4Institute of Precision Medicine, College of Medicine, National Sun Yat-sen University, Kaohsiung City 804, Taiwan

**Keywords:** bone scan, intercostal muscle invasion, lung metastasis, myxofibrosarcoma

## Abstract

Myxofibrosarcoma is a type of soft tissue sarcoma, predominantly characterized by a high propensity for local recurrence, albeit demonstrating a relatively diminished risk for distant metastasis. Its prevalence is notably higher in elderly patients. Here, we present a case of a 73-year-old woman diagnosed with Myxofibrosarcoma. She was subjected to a whole-body bone scan using ^99m^Tc-methylene diphosphonate (MDP) to survey potential bony metastasis. It revealed marked MDP accumulation with peripheral soft tissue uptake in the right lateral chest region of this patient. This imaging phenotype could potentially be attributed to the augmented vascularity within the tumor, a finding that was prominently displayed in this particular case.

**Figure 1 diagnostics-14-00857-f001:**
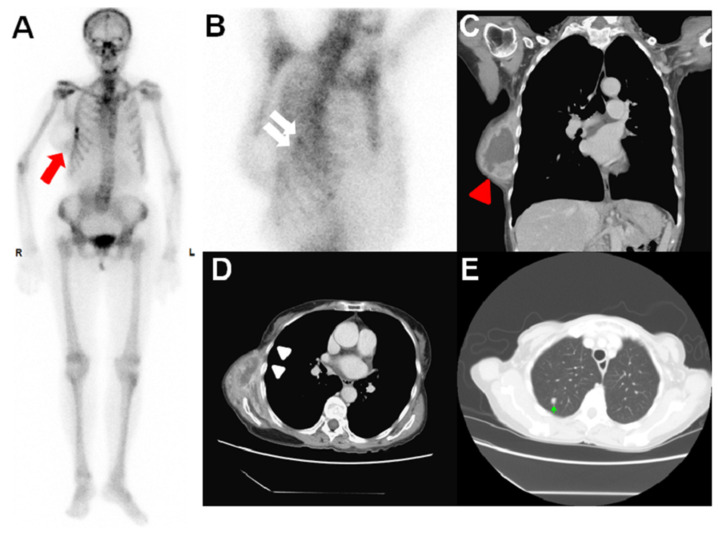
A 73-year-old woman presented with an enlarging mass over the right posterior and lateral upper trunk regions, and also complained of right upper limb weakness. An incisional biopsy of the tumor was performed, and the pathology report identified the mass as a high-grade myxofibrosarcoma (MFS). Given her symptoms, a ^99m^Tc-methylene diphosphonate (MDP) whole-body bone scan (WBBS) was arranged by a physician to survey for bony metastasis. The scan revealed a protruding soft tissue lesion in the lateral right chest region, with increased MDP uptake in the inferior aspect ((**A**), red arrow), and two MDP-avid foci in the area of the right 5th-6th ribs ((**B**), white arrows). A malposition of the right kidney was incidentally found. A chest CT scan revealed a large 6 × 9 × 12 cm protruding mass at the right chest wall with peripheral contrast enhancement with a central cystic-like component/multilobulated configuration ((**C**), red arrowhead), corresponding to the MDP uptake seen on the WBBS. It also shows invasion of the right 5th-6th intercostal muscles, but no invasion of the right 5th-6th ribs, contrary to what the WBBS was suggesting ((**D**), white arrowheads). There are no obvious osteolytic or osteoblastic lesions within the scan field of the CT. However, lung metastasis is also noted on the chest CT ((**E**), green arrow), leading to the arrangement of neo-adjuvant chemotherapy as the initial treatment. After six months, disease progression was noted, with multiple liver metastases. MFS typically arises in elderly patients, most commonly in the lower extremities, followed by the trunk. It is known to have a higher local recurrence rate (about 15~48%) but a relatively lower risk of distant metastases (near 10%) among soft-tissue tumors [1]. The treatment of MFS may involve a combination of surgical resection, radiation, chemotherapy, or targeted therapy, depending on the tumor status [2]. MRI or CT can play a crucial role in diagnosis and preoperative planning. In the literature, MFS has been reported for the detection of local tumor recurrence or metastasis using thallium-201 or FDG PET [3,4,5,6]. In our case, the MDP-avid tumor uptake and peripheral contrast enhancement/multilobulated configuration of the tumor may be related to an abundance of blood flow to the tumor. The WBBS also showed MDP-avid intercostal muscle invasion. A published article on MRI noted a multilobulated configuration of soft-tissue sarcoma, which correlates with the malignancy grade [7]. This configuration may be compatible with the biopsy results of this case. This case exemplifies the critical role of multimodal imaging in the diagnosis of myxofibrosarcoma, particularly in assessing tumor vascularity and morphology. It not only contributes to the existing literature by detailing a less common site and presentation of myxofibrosarcoma but also underscores the importance of an integrated approach to oncologic imaging in planning further treatments.

## Data Availability

The original data presented in the study are available on request from the corresponding author.

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
