# Peer review of "Myxofibrosarcoma of the Chest Wall Detected on 99mTc-MDP Whole-Body Bone Scan"

_diagnostics, 2024, doi:10.3390/diagnostics14080857_

Round 1

Reviewer 1 Report

Comments and Suggestions for Authors

Nice case presentation. However, do I understand correctly that the incisional biopsy was performed before any imaging? In my institution, I am used that biopsy is performed after imaging to make sure the hemorrhage from the biopsy does not conflict with the tumor characteristics. 
Also, did you consider 18FDG-PET or whole body CT for metastasis detection? 

Author Response

Dear reviewer:

Thank you the comments and suggestions. Please see the attached file for reply. 

Reviewer 2 Report

Comments and Suggestions for Authors

P1, line 4, Myxofibrosarcoma is one word, so it doesn't seem necessary to use an abbreviation as well as full manuscript.

P1, line 5-6 & P2, line 29-30, Please mention reported rates for local recurrence, distant metastasis, and prevalence.

P1, line 8-9, The location of the MDP accumulation is missing.

P1, line 23, I think a footnote called arrowhead would be more appropriate than a footnote called triangle for Figure C.

P2. I thick it would be good to summarize the clinical impact that can be informed from this case in the last sentence at the end.

P1~2, figure legend, The figure legend should be written in the present continuous tense. Please edit accordingly.

P1~2, figure, It would be helpful if the CT finding of myxofibrosarcoma could be further described. Also If you have MRI scans, please attach that as well.

Author Response

(The authors gave the same response as above.)

Reviewer 3 Report

Comments and Suggestions for Authors

I just have one minor request. As I see from the images, the tumor should have a "multilobulated configuration". Please add this term and please use the following paper for citing: https://doi.org/10.2478/raon-2021-0007

Comments on the Quality of English Language

Good.

Author Response

(The authors gave the same response as above.)

Round 2

Reviewer 2 Report

Comments and Suggestions for Authors

I have confirmed that my request has been successfully reflected in revised version.